



# Estimating cloud condensation nuclei concentrations from CALIPSO lidar measurements

Goutam Choudhury[1] and Matthias Tesche[1]

[1]Leipzig Institute for Meteorology (LIM), Leipzig University, Stephanstrasse 3, 04103 Leipzig, Germany

**Correspondence:** Goutam Choudhury (goutam.choudhury@uni-leipzig.de)

**Abstract.** We present a novel methodology to estimate cloud condensation nuclei (CCN) concentrations from spaceborne CALIPSO lidar measurements. The algorithm utilizes (i) the CALIPSO-derived backscatter and extinction coefficient, depolarization ratio, and aerosol subtype information, (ii) the normalized volume size distributions and refractive indices from the CALIPSO aerosol model, and (iii) the MOPSMAP optical modelling package. For each CALIPSO height bin, we first select the aerosol-type specific size distribution and then adjust it to reproduce the extinction coefficient derived from the CALIPSO retrieval. The scaled size distribution is integrated to estimate the aerosol number concentration which is then used in the CCN parameterizations to calculate CCN concentrations at different supersaturations. To account for the hygroscopicity of continental and marine aerosols, we use the kappa parameterization and correct the size distributions before the scaling step. We have studied the sensitivity of the thus derived CCN concentration to the effect of variations of the initial size distributions. It is found that the uncertainty associated with the algorithm can range between a factor of 2 and 3. We have also compared our results with the POLIPHON and found comparable results for extinction coefficients larger than 0.05 km$^{-1}$. An initial application to a case with coincident airborne in-situ measurements for independent validation shows promising results and illustrates the potential of CALIPSO for constructing a global height-resolved CCN climatology.

## 1 Introduction

Aerosol particles act as cloud condensation nuclei (CCN) and ice nucleating particles (INP) and provide a surface for the condensation of atmospheric water vapour to form cloud droplets. The physical and chemical properties of such particles not only affect the cloud micro and macro-physical properties, but also cloud development, lifetime, and the associated precipitation (Rosenfeld et al., 2014; Fan et al., 2016; Choudhury et al., 2019). The rapid adjustments in clouds resulting from aerosol-cloud interactions (ACI) are not well understood and still remains the largest source of spread in global climate projections (Masson-Delmotte et al., 2021). This challenge has motivated the scientific community to study ACI by using data from in-situ and satellite measurements as well as by means of modelling and simulations.

Satellites provide long term global coverage that enables ACI studies with constrained meteorology and cloud regimes (Oreopoulos et al., 2017; Douglas and L'Ecuyer, 2019; Jia et al., 2021). Satellite-based ACI studies relate cloud parameters (cloud reflectivity or albedo, cloud optical depth, cloud fraction, cloud drop effective radius, liquid water path), aerosol properties (aerosol optical depth (AOD), angstrom exponent(AE), aerosol index (AI)), and the precipitation pattern, to understand the





underlying mechanisms (Quaas et al., 2008; Gryspeerdt and Stier, 2012; McCoy et al., 2017; Kant et al., 2019; Liu et al., 2020; Choudhury et al., 2020; Quaas et al., 2020). Shinozuka et al. (2015) suggest that the satellite-derived AOD or AI, being a column integrated product, may not be the appropriate proxies for cloud-relevant CCN particles that usually lie close to the cloud base. Stier (2016) found low correlation (< 0.5) between the ECHAM-HAM model simulated AOD and CCN

concentration near the cloud base, and suggested the use of vertically resolved measurements from spaceborne lidars for ACI studies. The PSML003_Ocean data included in the MODIS (Moderate Resolution Imaging Spectroradiometer) ocean product gives the aerosol number concentration with a radius greater than or equal to 30 nm or $n_{30}$ (Remer et al., 2005, Appendix B). This product is formed by matching the spectral radiance measured by MODIS to the radiance estimated from a combination of the microphysical properties (size distributions and refractive indices) of 9 aerosol types. However, the column integrated

$n_{30}$ is proportional to the AOD and may not represent the atmospheric CCN particles located close to the cloud base altitudes (Shinozuka et al., 2015).

   Lidar provides height resolved aerosol optical properties which are crucial to study vertically co-located aerosols and clouds (Costantino and Bréon, 2013). Mamouri and Ansmann (2015) for the first time presented the Polarization Lidar Photometer Networking (POLIPHON) technique to estimate INP concentrations from lidar derived extinction coefficient for desert dust

aerosols. The algorithm first converts the extinction coefficient to aerosol number concentration with radius > 280 nm ($n_{280}$) by using conversion factors derived from the Aerosol Robotic Network (AERONET) correlation study. The INP concentration is then calculated from $n_{280}$ using the parameterizations from DeMott et al. (2010, 2015). Mamouri and Ansmann (2016) further extend the methodology for estimating CCN concentrations at different supersaturation from lidar-derived extinction coefficient for dust, continental and marine aerosols, and more recently for aged and fresh smoke aerosols (Ansmann et al.,

2021). The POLIPHON technique to estimate CCN and INP concentrations is not only limited to ground-based lidars but can also be applied to the spaceborne lidar CALIOP (Cloud-Aerosol Lidar with Orthogonal Polarization) aboard CALIPSO (Cloud-Aerosol Lidar and Infrared Pathfinder Satellite Observations) polar-orbiting satellite (Marinou et al., 2019; Georgoulias et al., 2020). This illustrates the potential of spaceborne lidar measurements to construct global 3D CCN and INP dataset.

   The CALIPSO aerosol model includes a set of normalized volume size distributions (NVSD) and refractive indices of 6

aerosol subtypes (Omar et al., 2009). Similar to the MODIS PSML003_Ocean algorithm, these microphysical properties along with the CALIPSO measured aerosol optical properties can be used to derive the cloud relevant aerosol number concentrations. In the present work, we utilize the CALIPSO aerosol model to calculate the extinction coefficient by using Mie scattering for spherical particles (continental and marine aerosols) and a combination of T-matrix and improved geometric optics method for non-spherical particles (dust aerosols). We then modify the NVSD by preserving its shape (mode radii and standard de-

viation remain constant) until a closure is achieved between the extinction coefficient inferred from CALIPSO measurements and derived through light-scattering calculations. We finally use the modified size distribution to compute the aerosol number concentration favourable to act as CCN by using the CCN paramterizations that correspond to different aerosol types (Mamouri and Ansmann, 2016). Further, we have performed sensitivity tests by varying the initial NVSD to quantify the uncertainty associated with the retrieval algorithm. We compare our results with the existing CCN retrieval algorithm POLIPHON for different

aerosol subtypes. Moreover, we present a case study where we have applied our algorithm to a CALIPSO overpass over Thes-


saloniki and compare it with the in-situ observations taken during the ACEMED-EUFAR (evaluation of CALIPSO's aerosol classification scheme over Eastern Mediterranean) campaign (Tsekeri et al., 2017). The data, optical modelling package used in this work, and a brief overview of the POLIPHON technique for retrieving CCN concentrations from lidars are described in Sections 2. Section 3 describes our CCN retrieval algorithm for spaceborne lidar. The sensitivity analysis and comparison
studies are presented in Section 4. We conclude the paper with a summary in Section 5.

## 2 Data and retrievals

### 2.1 CALIPSO

CALIPSO is a sun-synchronous polar-orbiting satellite launched on April 28, 2006, as a part of the afternoon or A-Train constellation (Winker et al., 2009). CALIOP is a polarization-sensitive lidar onboard CALIPSO that measures profiles of
aerosol and cloud properties from an elevation of 30 km above mean sea level to the surface. The CALIPSO algorithm classifies the measured signal into aerosols, clouds, clear air, and surface, and assigns a subtype to the detected aerosol signals (Omar et al., 2009). CALIPSO has a set of lidar ratios associated with each aerosol subtype. These lidar ratios are used in the CALIPSO retrieval algorithm to estimate the aerosol extinction and backscatter coefficient. In this work, we use the CALIPSO version 4.20 level-2 aerosol profile product with a uniform horizontal resolution of 5 km. Because of CALIPSO's data averaging
scheme, the vertical resolution of aerosol profile data varies with altitude. It is 60 m in between altitudes of 20 km to -0.5 km, and 180 m above 20 km. We use the profiles of aerosol extinction coefficient, backscatter coefficient, and depolarization ratio measured at 532 nm, and the aerosol subtype information in the CCN retrieval algorithm. We also use the relative humidity profiles included in the CALIPSO data product, obtained by the Global modelling and Assimilation Office Data Assimilation System.
The CALIPSO version 2 aerosol types include dust, smoke, clean continental, polluted continental, clean marine, and polluted dust. The microphysical properties of these six aerosol subtypes constitute the CALIPSO aerosol model (CAMel). The lidar ratios used in the retrieval of extinction coefficient for each aerosol type were modelled using these microphysical properties. Of the six aerosol subtypes, the properties of smoke, polluted continental, and polluted dust were obtained directly from a cluster analysis of long term cloud screened AERONET measurements (Omar et al., 2005). The dust model was derived from
Kalashnikova and Sokolik (2002) and the clean marine model was derived from the dry measurements taken during Shoreline Environment Aerosol Study (SEAS) campaign (Masonis et al., 2003; Clarke et al., 2003). The clean continental model was formed by adjusting the properties of the background continental aerosol cluster from Omar et al. (2005) to measurements of Anderson et al. (2000). The aerosol model has evolved with time. In version 4, a new aerosol subtype namely the dusty marine (dust + marine) was introduced. Further, the polluted continental and smoke subtypes were renamed to polluted con-
tinental/smoke and elevated smoke, respectively (Kim et al., 2018). The lidar ratios were also modified leading to an increase in mean AOD by 52% (40%) for nighttime (daytime) retrievals, making it more comparable with MODIS derived AOD. In our algorithm, we use the microphysical properties of 5 aerosol subtypes namely marine, dust, polluted continental/smoke, clean continental, and elevated smoke. Since the changes in lidar ratio from version 2 to version 4 are minor ($\leq 1\%$) for all





aerosol types except for clean continental (51%), we believe the aerosol models can still be used in our algorithm. However,

for the case of clean continental aerosol subtype, further study is required to estimate the effect of change in lidar ratio on

its microphysical properties. Having said that, we do not exclude it from our analysis for the completeness of our algorithm,

leaving a scope of future validation study to examine its applicability in estimating the CCN concentrations from CALIPSO.

## 2.2 MOPSMAP package

Modelled optical properties of ensembles of aerosol particles (MOPSMAP) package provides the aerosol optical properties of

arbitrary, randomly oriented spherical or spheroidal particle ensembles for size parameter ranging up to 1000 and refractive

index range of [0.1, 3.0] and [0, 2.2] for real and imaginary parts, respectively (Gasteiger and Wiegner, 2018). It includes

a data set of pre-calculated aerosol optical properties and a Fortran program which estimates the properties of user-defined

aerosol ensembles. The optical properties of spherical particles are modelled using Mie scattering. While for spheroids, based

on the aerosol size parameter, MOPSMAP uses a combination of the T-matrix method and improved geometric optics method.

MOPSMAP has been used to simulate the optical properties of different aerosol types such as mineral (silica and alumina) and

ash aerosols (Jiang et al., 2021), and Martian dust aerosols (Chen-Chen et al., 2021). We apply the MOPSMAP package to

model the aerosol extinction coefficient of different aerosol subtypes with the bimodal lognormal volume size distributions and

refractive indices from the CAMel. The details of the MOPSMAP input parameters are discussed in the methodology section.

## 2.3 POLIPHON

The POLIPHON technique enables the retrieval of aerosol number concentration by combining the ability of polarization

lidar to measure aerosol type-specific optical properties with long-term AERONET measurements of aerosol microphysical

properties and AOD (Mamouri and Ansmann, 2015, 2016). It converts the lidar derived extinction coefficient ($\alpha$) to number

concentration of aerosols with a dry radius greater than 100 nm ($n_{100,\mathrm{dry}}$) for dust aerosols and greater than 50 nm ($n_{50,\mathrm{dry}}$)

for marine and continental aerosols as

$$n_{j,\mathrm{dry}} = C \cdot \alpha^x(z), \tag{1}$$

where $n_{j,\mathrm{dry}}$ represents the total aerosol concentration with dry radius greater than $j$ nm, $C$ is the conversion factor and $x$ is the

extinction exponent. The value of $j$ is 50 nm for continental and marine aerosols, and 100 nm for dust aerosols. The constants

$C$ and $x$ are calculated from the $\log n_{j,\mathrm{dry}}$-$\log\alpha$ regression analysis of AERONET measurements and their values used in this

work are listed in Table 1.

The CCN concentration at a certain supersaturation is estimated from the aerosol number concentration as

$$n_{\mathrm{CCN}} = f_{\mathrm{ss}} \cdot n_{j,\mathrm{dry}}, \tag{2}$$

where $f_{\mathrm{ss}}$ = 1.0, 1.35, and 1.7 for supersaturations of 0.15%, 0.25%, and 0.40%, respectively. In this study, we use the conversion factors and extinction exponents for continental and marine aerosols from Mamouri and Ansmann (2016). For dust



aerosols, we use the globally averaged values as suggested by Ansmann et al. (2019) for application to satellite data. For

smoke aerosols, we use the aged smoke conversion factor and extinction exponent values from Ansmann et al. (2021).

## 3   Methodology

This section describes the algorithm used in the present work to derive CCN concentration from the CALIPSO measured profiles of extinction coefficient, backscatter coefficient, depolarization ratio, and aerosol subtype information. For continental and marine aerosols, particles with a dry radius greater than 50 nm forms the favourable CCN reservoir (Mamouri and Ansmann,

2016). While for desert dust aerosols, the minimum radius limit is 100 nm. The particle concentrations as $n_{50,\mathrm{dry}}$ (marine and continental) and $n_{100,\mathrm{dry}}$ (dust) are the input needed in the parameterizations to compute CCN concentrations at different supersaturation (Eq. 2). Thus, to estimate the CCN concentrations from CALIPSO, we first need to compute $n_{50,\mathrm{dry}}$ and $n_{100,\mathrm{dry}}$. This is done by adjusting the normalized size distributions from the CAMel as per the measured CALIPSO extinction coefficient.

### 3.1   Aerosol size distribution

The remote sensing of aerosol number concentration requires an initial assumption of aerosol microphysical properties (size distribution and refractive index). For instance, the MODIS algorithm over the ocean uses a combinations of 9 predefined aerosol size distributions and refractive indices, and selects the one for which the difference in the measured and modelled radiance is minimum (Appendix B of Remer et al. (2005)). In our study, we use the aerosol microphysical properties from

CAMel and adopt a two-step algorithm to derive the aerosol size distribution: (i) select the appropriate initial normalized volume size distribution and refractive index, and (ii) scale the size distribution as per the CALIPSO measured extinction coefficient. In contrast to MODIS, the aerosol type in CALIPSO is set prior to the computation of the extinction coefficient. This eases the selection of initial aerosol microphysics, which can now be done directly from the CAMel as per the aerosol subtype information included in the CALIPSO retrieval.

The next step is to scale the NVSD as per the CALIPSO measured extinction. The extinction coefficient ($\alpha$) for a certain incident wavelength can be described as

$$\alpha = \int\limits_{r_{\min}}^{r_{\max}} \frac{K_\alpha(m,r)}{V(r)} \cdot \frac{dV(r)}{d\ln r} \cdot d\ln r, \tag{3}$$

where $r$ is the particle radius, $V(r)$ is the volume of the particle with radius $r$ and $K_\alpha$ is the extinction cross-section which is a function of the complex refractive index ($m$) and $r$. $dV(r)/d\ln r$ is the log-normal volume size distribution which for a

bimodal case can be given by

$$\frac{dV(r)}{d\ln r} = V_{\mathrm{t}} \cdot \sum_{i=1}^{2} \frac{\nu_i}{\sqrt{2\pi}\ln\sigma_i} \exp\left(\frac{-(\ln r - \ln\mu_i)^2}{2\ln\sigma_i{}^2}\right). \tag{4}$$





Here, $\nu_i$, $\sigma_i$, and $\mu_i$ are the volume fractions, geometric standard deviations, and geometric mean radii of $i^{\text{th}}$ mode, respectively. $V_{\text{t}}$ is the total volume of the size distribution. The above size distribution is normalized when $V_{\text{t}} = 1$. Substituting Eq. (4) in Eq. (3), we get

$$\alpha = V_{\text{t}} \cdot \int_{r_{\min}}^{r_{\max}} \frac{K_\alpha(m,r)}{V(r)} \cdot \sum_{i=1}^{2} \frac{\nu_i}{\sqrt{2\pi}\ln\sigma_i} \exp\left(\frac{-(\ln r - \ln\mu_i)^2}{2\ln\sigma_i{}^2}\right) \cdot d\ln r. \tag{5}$$

Thus, the extinction coefficient is a function of the size distribution parameters ($V_{\text{t}}$, $\nu_i$, $\sigma_i$ and $\mu_i$) and the extinction cross section ($K_\alpha$). Out of these parameters, under ideal conditions, only $V_{\text{t}}$ is an extensive property, while the rest are intensive and independent of aerosol amount or concentration (Omar et al., 2005). Eq. (5) can be simplified to

$$\alpha = V_{\text{t}} \cdot \alpha_{\text{n}}. \tag{6}$$

Where $\alpha_{\text{n}}$ is the normalized extinction coefficient corresponding to the NVSD. If we consider $\alpha$ as the CALIPSO measured extinction, $V_{\text{t}}$ would be the scaling factor for the NVSD to compute the actual aerosol size distribution. From Eq. (6), we can compute $V_{\text{t}}$ if the value of $\alpha_{\text{n}}$ is known.

We estimate $\alpha_{\text{n}}$ for each aerosol subtype by using the NVSDs and refractive indices from CAMel as input to the MOPSMAP optical modelling package. In the MOPSMAP input, we consider dust as spheroids and use the axis ratio distribution from
Dubovik et al. (2006) (also used in the AERONET inversion). Other aerosol subtypes are considered spheres. We then compute $V_{\text{t}}$ from the ratio of $\alpha$ and $\alpha_{\text{n}}$ (Eq. 6). On multiplying $V_{\text{t}}$ with the NVSD, we get the final scaled aerosol size distribution. Since the algorithm principally relies on the optical modelling of CALIPSO aerosol microphysics, we hereafter refer to it as OMCAM.

## 3.2 Aerosol hygroscopicity

The hygroscopic aerosol particles in the atmosphere can uptake water and grow in moist conditions. The hygroscopic growth needs to be accounted for before deriving the aerosol size distributions discussed in the previous section. We consider continental (clean continental, polluted continental/smoke and elevated smoke) and marine aerosols as hygroscopic. We assume dust aerosols to be hydrophobic in accordance with previous studies (Mamouri and Ansmann, 2016; Ansmann et al., 2019). The hygroscopicity correction can be either applied to the ambient extinction coefficient measured by CALIPSO or to the
initial NVSD in the retrieval algorithm. We consider the latter approach and modify the initial NVSD before modelling the extinction coefficient. There is an inbuilt functionality in the MOPSMAP package to account for the hygroscopicity using the Kappa parametrization scheme (Petters and Kreidenweis, 2007; Zieger et al., 2013) as

$$\frac{r_{\text{wet}}(\text{RH})}{r_{\text{dry}}} = \left(1 + \kappa \cdot \frac{\text{RH}}{100 - \text{RH}}\right)^{\frac{1}{3}}, \tag{7}$$

where RH is the relative humidity and $\kappa$ is the hygroscopic growth parameter. The $r_{\min}$, $r_{\max}$, and $\mu$ of the log normal size
distribution (Eq. 5) are multiplied with this ratio whereas the standard deviation ($\sigma$) remains unchanged. The refractive index of the hygroscopic aerosol is also modified following the volume weighting rule (Gasteiger and Wiegner, 2018). The $\kappa$ value





is set to be 0.3 for continental and 0.7 for marine aerosols. The values are global averages and are suggested by Andreae and Rosenfeld (2008).

### 3.3 CCN parametrizations

We use the parameterizations listed in Mamouri and Ansmann (2016) to estimate CCN concentrations from the dry aerosol number concentration. The final scaled aerosol volume size distribution obtained from the scaling procedure is first converted to number size distribution. The number size distribution is integrated starting at 50 or 100 nm to compute $n_{50,\mathrm{dry}}$ or $n_{100,\mathrm{dry}}$ depending on the aerosol type. Finally, substituting the values in Eq. (2) results in the required CCN concentration at different supersaturations.

### 3.4 Application of OMCAM to CALIPSO retrieval

Figure 1 outlines the OMCAM retrieval algorithm for estimating CCN concentrations from CALIPSO measurements. In order to apply the OMCAM algorithm to CALIPSO level 2 version 4.20 data, we first start by preprocessing the dataset. To begin with, we apply all the quality filters listed in Tackett et al. (2018, Table 1). The CALIPSO aerosol typing algorithm consists of dust mixtures (dusty marine and polluted dust). In such a case, we separate the dust and non-dust extinction coefficients by 195 using the methodology given in Tesche et al. (2009). This is a rather simple and accepted dust separation technique also used by Mamouri and Ansmann (2015, 2016) for lidar based CCN retrieval. It uses the particle depolarization ratio ($\delta_{\mathrm{p}}$) to separate the particle backscatter coefficient ($\beta_{\mathrm{p}}$) into dust ($\beta_{\mathrm{d}}$) and non-dust ($\beta_{\mathrm{nd}}$) contributions. $\beta_{\mathrm{d}}$ can be calculated as

$$\beta_{\mathrm{d}} = \beta_{\mathrm{p}} \frac{(\delta_{\mathrm{p}} - \delta_2)(1 + \delta_1)}{(\delta_1 - \delta_2)(1 + \delta_{\mathrm{p}})}, \tag{8}$$

Where the values of $\delta_1$ and $\delta_2$ are 0.31 and 0.05, respectively. The aerosol mixture is assumed to be pure dust (non-dust) when 200 $\delta_{\mathrm{p}} > 0.31$ ($< 0.05$). When $0.05 \leq \delta_{\mathrm{p}} \leq 0.31$, we first estimate $\beta_{\mathrm{d}}$ from Eq. (8) and then calculate $\beta_{\mathrm{nd}}$ by subtracting $\beta_{\mathrm{d}}$ from $\beta_{\mathrm{p}}$. We compute the dust and non-dust extinction coefficient by multiplying the backscatter coefficient with the respective lidar ratio. The lidar ratios of dust, polluted continental and clean marine aerosol subtypes are taken from Kim et al. (2018) and are equal to 44, 70, and 23, respectively. The extinction coefficient of polluted dust is separated into polluted continental/smoke and dust, while that of dusty marine is separated into dust and marine contributions. Finally, the extinction coefficient, relative 205 humidity, and aerosol subtype information are passed to the CCN retrieval algorithm.

In the CCN retrieval part, we first select the normalized size distribution and refractive index as per the aerosol subtype and modify them as per the RH value so as to account for the hygroscopicity of aerosols. In the next step, we model the extinction coefficient using the MOPSMAP package and calculate $V_{\mathrm{t}}$ from Eq. (6). Multiplying $V_{\mathrm{t}}$ with the initial dry normalized size distribution gives the final dry aerosol size distribution which is used in the CCN parameterizations (Eq. 2) to estimate the CCN 210 concentrations at different supersaturation values. This methodology is applied to every bin of the CALIPSO profile. In the case of dust mixtures, the separated dust and non-dust extinction coefficients are passed through the CCN retrieval algorithm individually, and the results are finally added to compute the net CCN concentration for that bin. It is worthwhile to note that this algorithm can in principle be used to derive INP concentration from CALIPSO measurements. This can be done by first





estimating $n_{250}$ from the modified size distribution (Section 4.1) and then using the INP parameterizations (DeMott et al., 2010, 2015) to estimate INP concentrations. However, in the present study, we limit our focus on retrieving the CCN concentrations.

## 4 Results

### 4.1 Sensitivity analysis

The performance of OMCAM in retrieving CCN concentrations primarily relies on the initial NVSD given in the CAMel.
The aerosol size distributions may change depending on the age and composition of aerosols (region and type dependent),
and the ambient meteorology. As most of the size distributions used in the CAMel are derived from cluster analysis of the long term AERONET measurements (see Section 2.2), they incorporate the errors associated with the AERONET inversion algorithm. Dubovik et al. (2000) found that the relative error in the AERONET-retrieved volume size distribution for dust, biomass burning, and water-soluble aerosols can go beyond 50% for both small ($r < 0.1$ μm) and large ($r > 7$ μm) particles. In order to account for such errors and natural variability, we analyzed the sensitivity of CCN concentrations to the initial
normalized size distributions considered in our retrieval algorithm.

For each aerosol subtype, the initial NVSD can be perturbed by changing the size distribution parameters such as the volume fractions ($\nu_{\mathrm{f}}$ & $\nu_{\mathrm{c}}$), geometric standard deviations ($\sigma_{\mathrm{f}}$ & $\sigma_{\mathrm{c}}$), and mean radii ($\mu_{\mathrm{f}}$ & $\mu_{\mathrm{c}}$) of fine and coarse modes. Since the sum of the volume fractions is unity, this leads to 5 independent size distribution parameters. We first study the individual effects of varying these parameters on the output $n_{j,\mathrm{dry}}$ ($j = 100$ for dust and 50 for other aerosol subtypes), as they are the
main input to the CCN parameterizations. Figure 2 depicts the effect of varying these size distribution parameters by $\pm 50\%$ on the $n_{j,\mathrm{dry}}$ relative to that of unperturbed size distributions from CAMel, for a preset $\alpha = 0.1$ km$^{-1}$ and RH = 0 for different aerosol subtypes. The results show fine mode as the primary contributor to the output aerosol number concentration. A certain change in the volume size distribution in the fine mode will have a larger impact on the number concentration compared to the coarse mode, as a much larger number of small particles is needed to produce the same change in volume. Out of the
5 parameters, $\mu_{\mathrm{f}}$ has the maximum effect ($\approx 800\%$) on the output number concentration followed by $\sigma_{\mathrm{f}}$ ($\approx 150\%$). This is because both $\mu_{\mathrm{f}}$ and $\sigma_{\mathrm{f}}$ modify the distribution of volume across different radii in the fine mode. Decreasing (increasing) $\mu_{\mathrm{f}}$ shifts the fine mode towards a smaller (larger) radius thereby resulting in a comparatively larger (smaller) number of particles for a constant fine mode volume. However, for dust, the effect is opposite when $\mu_{\mathrm{f}}$ is decreased. This is because the minimum cut-off radius for dust is set to be 100 nm and the fine mode moves out of this limit when $\mu_{\mathrm{f}}$ is reduced leading to a decrease in
the output number concentration. Increasing (decreasing) $\sigma_{\mathrm{f}}$ leads to an increase (decrease) in the fraction of smaller particles within the fine mode. This results in an increase (decrease) in the output number concentration for all aerosol subtypes except dust. The output number concentration is comparatively less sensitive to coarse mode parameters ($\mu_{\mathrm{c}}$ & $\sigma_{\mathrm{c}}$), as they contribute primarily to the optical properties of the aerosol volume rather than the number concentration. When we change the value of $\alpha$, the aerosol number concentration scales as per the ratio between $\alpha$ and $\alpha_{\mathrm{n}}$, resulting in no change in the relative $n_{100,\mathrm{dry}}$
and $n_{50,\mathrm{dry}}$.





The size distributions formed by varying the size distribution parameters separately may not be sufficient enough to capture the natural variability. Thus to imitate the natural variability in a better way, we further consider combinations of the variations of all the parameters. We don't expect extreme shifts in the size distribution parameters as well. For instance, reducing $\mu_{\mathrm{f}}$ by 50% results in abnormal size distributions with 30%-50% of the fine mode moving out of the AERONET size limits (0.05 $\leq r \leq 15$ µm). Therefore, in order to exclude the non-physical size distributions, we limit the variations of the parameters in terms of the actual volume size distributions. To implement these constraints, we first vary the size distribution parameters linearly with a uniform spacing of 0.01 and then consider all possible combinations of the variations. The NVSDs generated from all the combinations forms the input NVSD set for the sensitivity analysis. We further fix the maximum limits of bimodal NVSD to ± 50% of the amplitude of each of its modes and do not consider the NVSDs that fall outside this domain in the sensitivity studies. The resulting input NVSD space for each aerosol type is shown by the shaded region of Figure 3. The maximum and minimum values of all the size distribution parameters considered in the sensitivity analysis are given in Table 2.

As we have kept a constant spacing for varying the size distribution parameters, the number of NVSD in the input space directly depends on the volume of particles present in each mode. While it is minimum for clean marine subtype because of its almost non-existent fine mode (which reduces the range of variation), it is maximum for polluted continental and elevated smoke subtypes. The output ensemble of number concentrations for an extinction coefficient of $0.1 \, \mathrm{km}^{-1}$ and relative humidity of 0% are shown in the violin plots of Figure 4. The percentiles of the output $n_{j,\mathrm{dry}}$ set are given in Table 3. The number concentration of the output ensemble is primarily dependent on the fine mode of the input size distributions. The variations in the output ensemble relative to the output from unperturbed NVSD from CAMel is minimum (about a factor of 1) for dust mainly because we only consider particles with a radius > 0.1 µm. For clean marine, the spread is about a factor of 2 ($95^{\mathrm{th}}$ percentile; 200%). However, for polluted continental and elevated smoke, the output ensemble is bi-modal. For the first mode, the values can go up to a factor of 1.5 for polluted continental and to around 1 for elevated smoke. The second mode is relatively small and is related to the size distributions whose fine mode mean radii are shifted to low values (extreme left in Figure 2). For this mode, the values can go up to a factor of 3 for polluted continental and 2.5 for elevated smoke. The largest spread in the output ensemble is found for clean continental ($95^{th}$ percentile; factor of 2.7). This might be because the bi-modality of the NVSD is not well defined for the clean continental aerosol subtype, thereby increasing the input space of variation. Neglecting the long tail of the distribution, we can assume the uncertainty to be about a factor of 2.

We have also estimated the effect of change in RH on the output ensemble of $n_{100,\mathrm{dry}}$ and $n_{50,\mathrm{dry}}$ (not shown). Increasing RH decreases the spread of the output ensemble slightly, with a significant decrease for RH > 90% except for dust which is assumed to be hydrophobic. At RH = 99%, the bi-modality of polluted continental and elevated smoke subtypes disappears. The variations in the relative number concentrations decrease to less than a factor of 2 for all subtypes. This might be a result of the decrease in the absolute number concentration, as the particle size increases with RH and fewer particles are needed to produce the same extinction. At a constant RH value, when $\alpha$ is modified, the output ensemble of aerosol number concentrations scales as per the ratio between $\alpha$ and $\alpha_{\mathrm{n}}$ resulting in no change in the relative $n_{100,\mathrm{dry}}$ and $n_{50,\mathrm{dry}}$ (not shown). To summarize, if we neglect the contributions of extreme shifts in the size distribution (i.e., the long tails in the violin plots) and consider the effect





of RH, we can assume the overall uncertainty in the retrieval algorithm due to the initial NVSD is likely to range between a factor of 1.5 and 2.5.

Uncertainties in the OMCAM algorithm can also arise from the uncertainty in the CALIPSO measurements, the CCN parametrization, and the hygroscopicity parametrization. The CALIPSO retrieved extinction coefficient can have an uncertainty

of up to 30% (Omar et al., 2009; Kim et al., 2018). The ability of aerosol to act as CCN depends on the composition, size, and atmospheric supersaturation value. In situations with complex aerosol mixtures and variable updraft velocity, the simple CCN parametrization developed by Mamouri and Ansmann (2016) may fail. The $\kappa$ values used to account for the hygroscopicity are global averages and may vary regionally depending on the aerosol source, composition and age. Moreover, the hydrophobic approximation for dust may not work for cases in which dust is coated/mixed with soluble aerosols (Mamouri and Ansmann,

2016). In such a case, dust aerosols with a dry radius > 50 nm can also act as CCN (Mamouri and Ansmann, 2016). Accounting for the mentioned possibilities, we assume that the overall uncertainty in our retrieval algorithm is likely to range between a factor of 2 and 3. It is comparable to the uncertainty of POLIPHON retrieval. However, OMCAM incorporates additional uncertainties due to the hygroscopicity correction. Studies have found that the conversion factors used in the POLIPHON technique for dust and smoke aerosols vary with the source region and the age of aerosols (Ansmann et al., 2019, 2021). Such

factors further increase the uncertainties associated with the retrieval algorithm when applied to satellite/global data sets.

## 4.2   Comparison with POLIPHON

In this section, we compare the CCN concentrations estimated from the OMCAM algorithm with that of the POLIPHON method (Mamouri and Ansmann, 2016). The ratio between the CCN concentrations estimated using POLIPHON ($CCN_{POLI}$) and OMCAM ($CCN_{OMCAM}$) algorithms for varying extinction coefficients at a supersaturation of 0.15% is shown in Figure 5.

The continental aerosols in POLIPHON represent a mixture of urban haze, biomass burning, road dust, and biological particles (Mamouri and Ansmann, 2016). Thus we compare it with the polluted continental aerosol subtype of CALIPSO. For continental aerosols, $CCN_{POLI}$ and $CCN_{OMCAM}$ are comparable, with the former being always larger than the latter. For smoke aerosols, both the algorithms yield similar values for $\alpha > 0.05$ km$^{-1}$. For $\alpha < 0.05$ km$^{-1}$, the POLIPHON values can be up to 2 times larger than that of OMCAM. The CCN concentrations estimated from both the algorithms yield similar results for dust as well,

with comparable values for $\alpha > 0.1$ km$^{-1}$ and increasing disparity for decreasing $\alpha$ below 0.05 km$^{-1}$. In the derivation of the conversion factors and extinction exponents in the POLIPHON method by regression analysis, the sample size for AOD < 0.05 is either zero for dust (Ansmann et al., 2019) or limited for smoke aerosols (Ansmann et al., 2021). It might be a reason behind the difference between the $CCN_{POLI}$ and $CCN_{OMCAM}$ for smoke and dust aerosols for $\alpha < 0.05$ km$^{-1}$. However, for the case of marine aerosols, the values estimated using POLIPHON are significantly larger than that of OMCAM (up to 6 times).

This may be either because of different instruments or sample size considered to derive the size distributions used in both the algorithms. POLIPHON conversion factors are estimated from the 7.5 years of measurements from the AERONET station located in Barbados (Mamouri and Ansmann, 2016), whereas the marine model used in OMCAM is derived from the in-situ measurements of sea-salt size distributions produced from breaking waves, taken during the SEAS experiment at Bellows Air Force Station, Oaahu, Hawaii in between 21-30 April 2000. Studies have found that the AERONET size distributions can





be significantly different from the in-situ measurements especially under high relative humidity conditions (Chauvigné et al., 2016; Schafer et al., 2019). Further studies involving type-specific comparison of both the aerosol number concentrations and the CCN concentrations with in-situ measurements are required to test the reliability of both algorithms.

## 4.3 Case study

In this section, we compare the profiles of aerosol number concentrations derived using the OMCAM and POLIPHON algo-
rithms with the in-situ observations taken during the ACEMED-EUFAR campaign (evaluation of CALIPSO's aerosol classi-fication scheme over Eastern Mediterranean). Specifically, we use the $n_{50,\text{dry}}$ concentrations estimated from the in-situ mea-surements taken on 9 September 2011 at 00:05-01:50 UTC over land and sea surface around Thessaloniki given in Tsekeri et al. (2017, Table 3,5) (hereafter referred to as T17). The airborne in-situ measurements coincide in space and time with the CALIPSO nighttime overpass at 00:40 UTC over Thessaloniki. Georgoulias et al. (2020) (hereafter written as G20) applied the
POLIPHON method to the overlapping CALIPSO measurements and estimated the CCN concentrations at a supersaturation of 0.15% ($n_{100,\text{dry}}$ for dust and $n_{50,\text{dry}}$ for continental and marine aerosols) for comparison with the in-situ measurements from T17. We apply the OMCAM algorithm to the same CALIPSO overpass and compute the $n_{50,\text{dry}}$ concentrations. The results are discussed as follows.

The profiles of CALIPSO measured extinction coefficient, aerosol subtype, and the $n_{50,\text{dry}}$ concentration calculated from the
OMCAM algorithm for the CALIPSO overpass over Thessaloniki on 9 September 2011, are shown in Figure 6. Over the land areas (latitude from 40.6°-41.2°N), CALIPSO aerosol typing algorithm identifies the presence of elevated smoke and polluted continental aerosols (Figure 6b). However, for retrieving the extinction coefficient for polluted continental aerosol layer, the lidar ratio was modified and, thus, is not considered in our present comparison (not shown). The presence of smoke over the land region was also identified by T17. The CALIPSO measured extinction coefficient over land is highly variable in space
ranging from 0.07 $\text{km}^{-1}$ to as high as $\approx 3\ \text{km}^{-1}$ in the proximity of cloud. The OMCAM estimated $n_{50,\text{dry}}$ correspondingly varies from 617 $\text{cm}^{-3}$ to 40000 $\text{cm}^{-3}$. Over the sea region (latitude from 40°-40.6°N), T17 detected the presence of elevated smoke plumes. This was not detected by the aerosol typing algorithm of earlier version-3 CALIPSO data used in T17. However, with the modifications of version 4 used in this work, CALIPSO successfully detects elevated smoke, marine, and dust aerosols with elevated smoke being the dominant one. The overall extinction over the sea area is less compared to land, with the values
ranging from 0.026 $\text{km}^{-1}$ to 0.36 $\text{km}^{-1}$. The corresponding OMCAM estimated $n_{50,\text{dry}}$ concentrations vary from 33 $\text{cm}^{-3}$ to 5000 $\text{cm}^{-3}$.

T17 estimated the $n_{50,\text{dry}}$ at different altitudes over the land region corresponding to two 5 km cloud-free segments of CALIPSO retrieval with latitudes in between 40.85°N and 40.95°N. The average $n_{50,\text{dry}}$ concentration estimated for the selected CALIPSO segments over land using OMCAM and POLIPHON (taken from G20) are plotted along with the in-situ
measurements from T17 in Figure 7a and the values are listed in Table 4. On average, when no hygroscopicity correction is applied, the OMCAM and POLIPHON overestimate the $n_{50,\text{dry}}$ concentration by 355% and 370%, respectively. A similar result from OMCAM and POLIPHON is expected given that elevated smoke was the dominant aerosol type over the land with extinction coefficient > 0.1 $\text{km}^{-1}$, for which both the algorithms yield a similar result (Figure 5). Upon accounting for the




hygroscopic growth, the overestimation decreases to 167% (130% for POLIPHON). Note that the RH-corrected POLIPHON
values in G20 are produced by using the in-situ dry to ambient extinction coefficient ratios (DAR) measured at different
RH values during the aircraft measurements (Tsekeri et al., 2017). In contrast to the overestimation over the land, both the
algorithms underestimate the $n_{50,dry}$ concentrations over the sea (Figure 7b). When we don't account for the hygroscopic
growth, both the OMCAM and POLIPHON algorithms underestimate the $n_{50,dry}$ concentration by 22% and 38%, respectively.
When the RH growth is corrected, the underestimation further increases to 40% and 52%, respectively. Similar to land region,
both the algorithms yield comparable results over the sea, as the dominant aerosol type is elevated smoke in both scenarios.

The $n_{50,dry}$ estimated over the land and sea region from the OMCAM and POLIPHON algorithms are comparable to each
other. The RH corrected POLIPHON values (using in-situ DAR measurements) are in good agreement with that of OMCAM
which uses kappa parametrization with globally averaged kappa values. Both the OMCAM and POLIPHON algorithms were
able to capture the pattern of altitudinal variations of $n_{50,dry}$ as observed by the in-situ measurements. However, the magnitudes
of $n_{50,dry}$ are overestimated by both the algorithms over the land by a factor of 1.5. Whereas over the sea region, the underes-
timation by both the algorithms is about a factor of 0.5. One of the intrinsic limitations of this comparison results from the vast
difference in measuring time scales of CALIPSO and the research aircraft. While for CALIPSO it is as small as 15 seconds,
it is around 2 hours for the aircraft. From Figure 6c, we can clearly see that the extinction coefficient along with the $n_{50,dry}$
concentrations is highly variable over the land region (ranging from 617 $cm^{-3}$ to 40000 $cm^{-3}$) compared to rather homoge-
neous concentrations over the sea. This might be the reason for such huge differences between in-situ and CALIPSO retrievals
over the land region. Moreover, only two cloud-free CALIPSO 5 km profiles are considered for the comparison over land,
which further increases the chances of disparity. Given the limited sample space, this comparison should not be considered as
validation but rather a demonstration of the capability for retrieving CCN concentrations from spaceborne lidar measurements.
A detailed study comparing the CALIPSO retrieved aerosol number and CCN concentrations with ground-based and aircraft
in-situ measurements is required to evaluate the reliability of OMCAM and POLIPHON algorithms in estimating the CCN
concentrations.

## 5   Summary and conclusions

We present the OMCAM algorithm to derive the height-resolved cloud relevant CCN concentrations from CALIPSO measure-
ments. The algorithm uses the normalized size distributions and refractive indices from CALIPSO aerosol models (Omar et
al., 2009) as an input to MOSPMAP to calculate the extinction coefficient. The size distributions are then scaled to reproduce
the CALIPSO measured extinction coefficient. In order to account for the hygroscopicity, we use $\kappa$ parametrization (Petters
and Kreidenweis, 2007), and modify the size distribution and the refractive index before the scaling step. We then estimate
the required aerosol number concentration by integrating the final scaled size distributions over the size ranges relevant for
different aerosol types. Utilizing the aerosol type-specific CCN parameterizations from the POLIPHON method (Mamouri and
Ansmann, 2016), we convert the aerosol number concentrations to cloud relevant CCN concentrations for different supersatu-
ration.





The OMCAM algorithm relies on the potentiality of the CALIPSO aerosol models to accurately describe the microphysical properties of the aerosol subtypes defined within the CALIPSO retrieval algorithm. We performed sensitivity tests by varying the normalized size distributions by up to $\pm 50\%$ of the amplitude of each mode and found that the uncertainty in the final aerosol number concentration ranges between a factor of 2 and 3.

We compared the CCN concentrations obtained from OMCAM with that of the POLIPHON method–the existing method for lidar-based CCN retrieval. For extinction coefficient $> 0.05$ km$^{-1}$, we found a good agreement for continental, dust, and smoke aerosols. However, as the extinction coefficient becomes smaller than $0.05$ km$^{-1}$, the difference increases with the POLIPHON values going as high as twice the OMCAM values. For marine aerosols, the CCN concentration derived using the POLIPHON method is always higher (4-6 times) than that of OMCAM.

For an initial evaluation of the OMCAM algorithm, we compared the thus obtained $n_{50,\mathrm{dry}}$ with in-situ measurements taken over the land and sea region around Thessaloniki during the ACEMED campaign (Tsekeri et al., 2017). For the retrievals over sea, we found that CALIPSO is underestimating the $n_{50,\mathrm{dry}}$ by about 40%. Over the land areas, however, CALIPSO overestimates $n_{50,\mathrm{dry}}$ by about 167%. The large discrepancies may be a result of the combination of highly variable $n_{50,\mathrm{dry}}$ over the land region and the instantaneous measurement by CALIPSO, in contrast to the in-situ measurement which were performed in a time period of 2 hours. All values remained within a factor of 2 which is in agreement with the estimated uncertainty. Moreover, the $n_{50,\mathrm{dry}}$ retrieved from CALIPSO using the OMCAM algorithm was comparable to that of POLIPHON (Georgoulias et al., 2020).

Our future goals include a comprehensive evaluation of the CALIPSO derived aerosol number and CCN concentrations with ground-based and airborne in-situ measurements. We will use the airborne Atmospheric Tomography Mission measurements of aerosol number concentration profiles from altitudes of 0.2 km to 12 km between the years 2016 and 2018 (Williamson et al., 2019) to access the quality of the respective parameter derived from CALIPSO. Furthermore, we will also compare the CALIPSO products with the long-term surface measurements of CCN and aerosol size distributions from 11 atmospheric observatories around the globe between 2006 and 2016 (Schmale et al., 2017). The comparison study will enable us to test the applicability of OMCAM and POLIPHON algorithms in the context of estimating aerosol number and CCN concentrations from spaceborne lidar measurements. Ultimately, we plan to apply the best performing algorithm to more than 15 years of CALIPSO data to construct a global height resolved CCN climatology. The data set when coupled with other satellite based global cloud related data or state of the art numerical models will help in improving our current understanding of the aerosol-cloud interactions.

The ability of CALIPSO not only in measuring vertically resolved aerosol optical properties but also being able to detect the responsible aerosol type has facilitated the retrieval of global 3D type-specific aerosol properties. We have described a novel methodology to retrieve cloud relevant CCN concentrations from CALIPSO measurements illustrating the potential of CALIPSO to produce 3D global CCN climatology for ACI studies and climate model evaluations, opening new gates for further validation of the algorithm against ground-based and airborne in-situ measurements.



*Data availability.* The CALIPSO level 2 v4.20 aerosol profile data product used in this work is available at https://doi.org/10.5067/CALIOP/
CALIPSO/LID_L2_05KMAPRO-STANDARD-V4-20 (CALIPSO, 2018)

.

*Author contributions.* MT conceptualized the study. GC developed the methodology under guidance of MT. GC performed the data analysis
and prepared the plots. GC and MT contributed equally to the interpretation of the data as well as to the preparation and revision of the
manuscript.

*Competing interests.* The authors declare that they have no conflict of interest.

*Acknowledgements.* We are grateful to Josef Gasteiger (University of Vienna, Vienna, Austria) for his help in compiling the MOPSMAP
package. We are thankful to Albert Ansmann (Leibniz Institute for Tropospheric Research, Leipzig, Germany) for fruitful discussion on the
overall algorithm. We thank the CALIPSO Science Team for providing the CALIPSO data. We thank the AERIS/ICARE Data and Services
Center for providing access to the CALIPSO data used in this study.

*Financial support.* This research has been supported by the Franco-German Fellowship Programme on Climate, Energy, and Earth System
Research (Make Our Planet Great Again – German Research Initiative, MOPGA-GRI, grant number 57429422) of the German Academic
Exchange Service (DAAD), funded by the German Ministry of Education and Research.



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



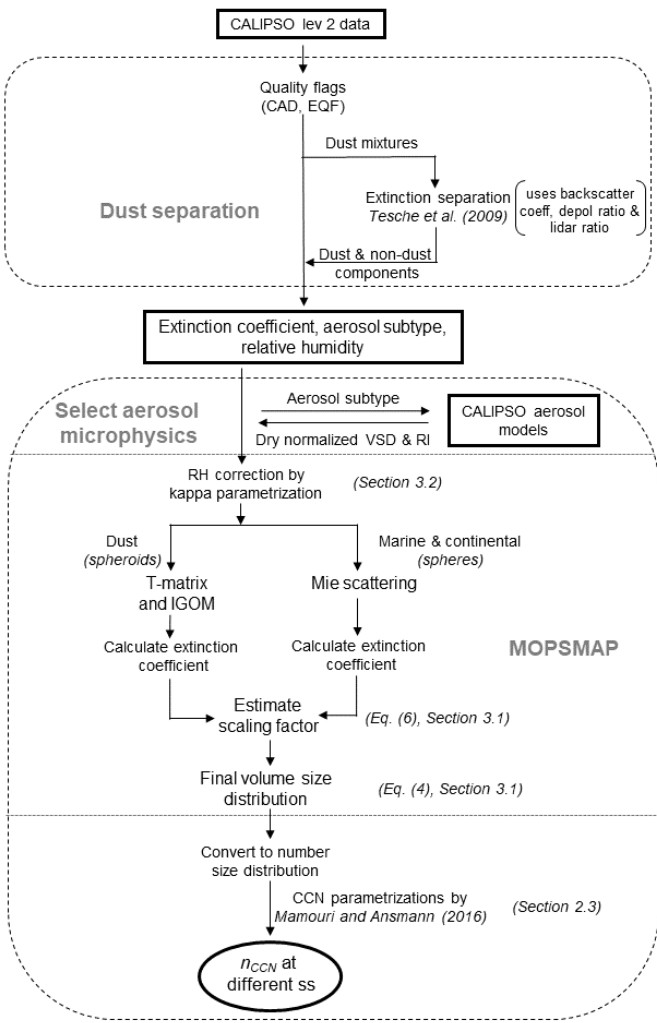

**Figure 1.** Flowchart of the OMCAM algorithm illustrating important steps involved in retrieving CCN concentrations from CALIPSO level 2 aerosol profile data. The upper part describes the pre-processing to infer information on the extinction coefficient, aerosol subtype, and the relative humidity. These parameters form the input to the CCN retrieval part which is outlined in the lower part. The chart also links to the used equations and the sections in which specific parts are discussed.

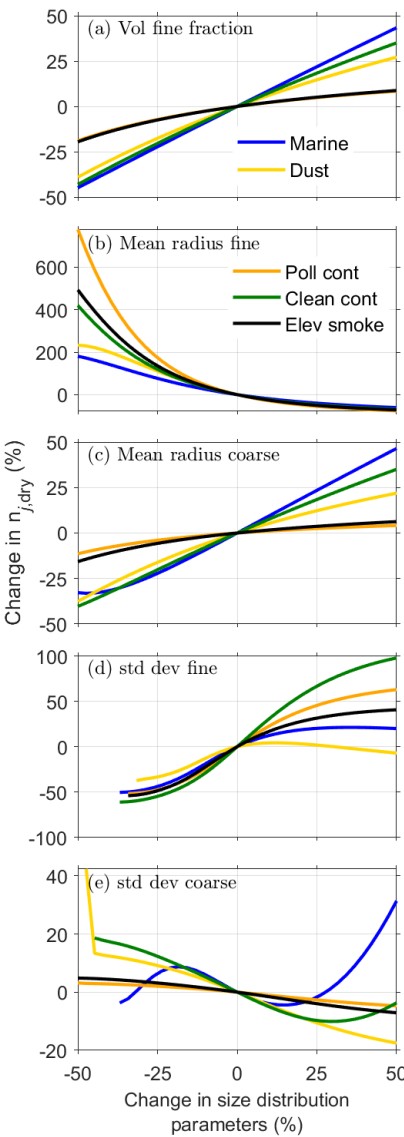

**Figure 2.** Sensitivity of $n_{j,\mathrm{dry}}$ ($j = 100$ for dust and 50 for other aerosol subtypes) to the size distribution parameters: volume fine fraction (a), mean radius fine (b), mean radius coarse (c), standard deviation fine (d), and standard deviation coarse (e). The x-axis represents perturbations in the size distribution parameters in percentage from their original values taken from CALIPSO aerosol model. Y-axis represents the corresponding percentage change in $n_{j,\mathrm{dry}}$ relative to that estimated from the unperturbed size distribution.



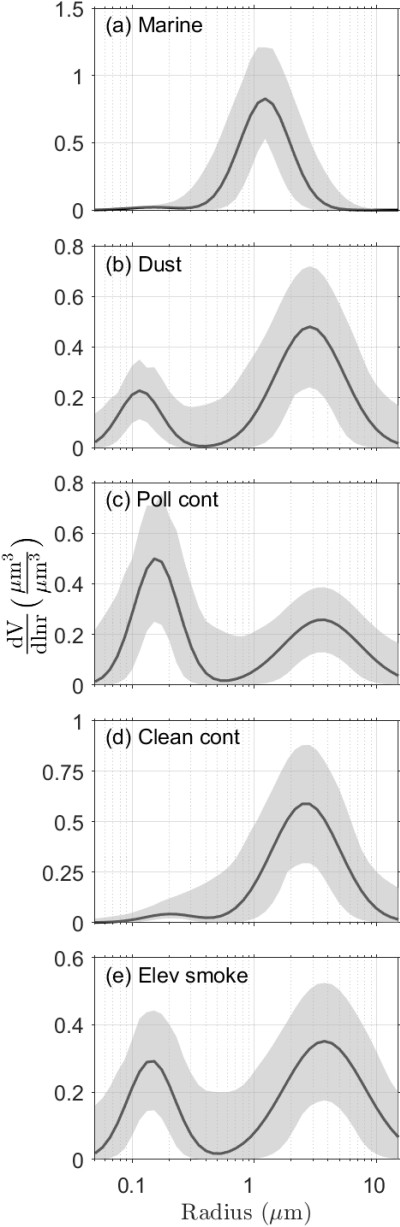

**Figure 3.** Normalized bi-modal log-normal volume size distributions for marine (a), dust (b), polluted continental (c), clean continental (d), and elevated smoke (e) aerosol subtypes adopted from the CALIPSO aerosol model. The shaded region represents the maximum and minimum limits of size distributions selected for the sensitivity analysis

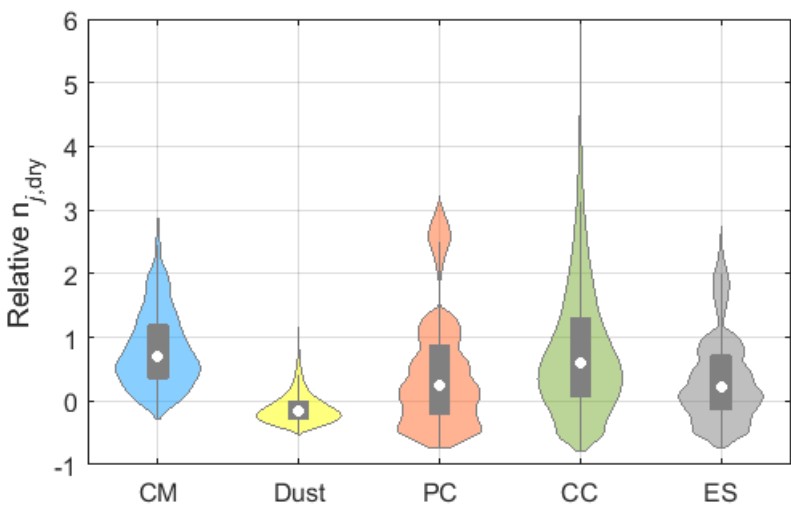

**Figure 4.** Violin plots for the output ensemble of $n_{j,\mathrm{dry}}$ ($j$ = 100 for dust and 50 for other aerosol subtypes) relative to that of unperturbed NVSD from CALIPSO aerosol model. The filled shape shows the probability density of the data (smoothed by non-parametric kernel density estimator) along the y axis, symmetric on either side representing a violin like shape. The box limits represent the first and third quartiles, the white circle inside the box is the median, and the ends of the grey line passing through the center of the box represent the adjacent values (data excluding outliers). Abbreviations: CM - clean marine, PC - polluted continental, CC - clean continental, and ES - elevated smoke.



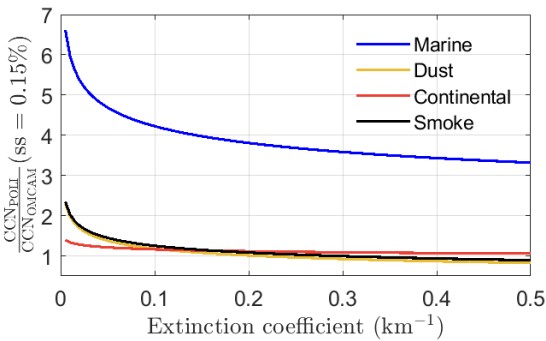

**Figure 5.** Ratio of $CCN_{ss=0.15}$ estimated from POLIPHON and OMCAM algorithms for varying extinction coefficient for marine, dust, continental, and smoke aerosol subtypes.



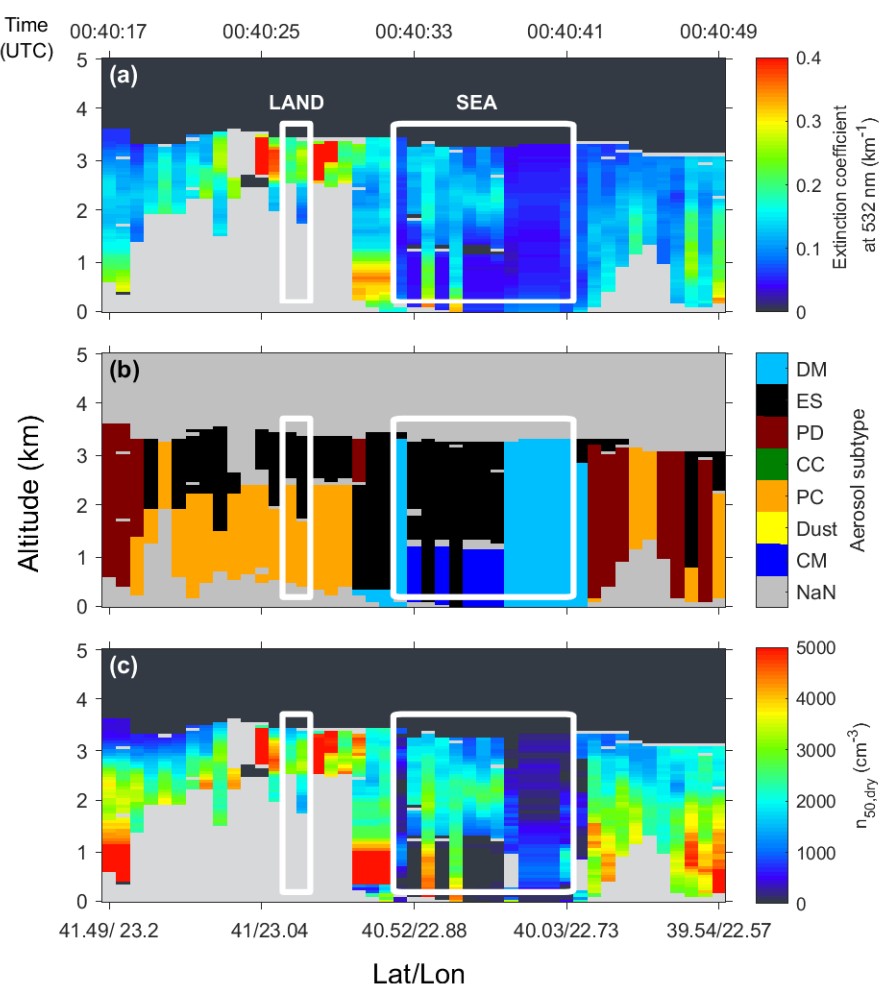

**Figure 6.** Plot of extinction coefficient (a), aerosol subtype mask (b), and the OMCAM estimated $n_{50,\mathrm{dry}}$ concentrations (c) for a CALIPSO overpass over Thessaloniki region on 9 September 2011. The white lines mark the land ($40.85° - 40.95°N$) and sea ($40° - 40.6°N$) regions for which the in-situ observations at different altitudes are provided by Tsekeri et al. (2017). The grey color represents invalid values (NAN). Abbreviations: CM - clean marine, PC - polluted continental, CC - clean continental, PD - polluted dust, ES - elevated smoke, and DM - dusty marine.



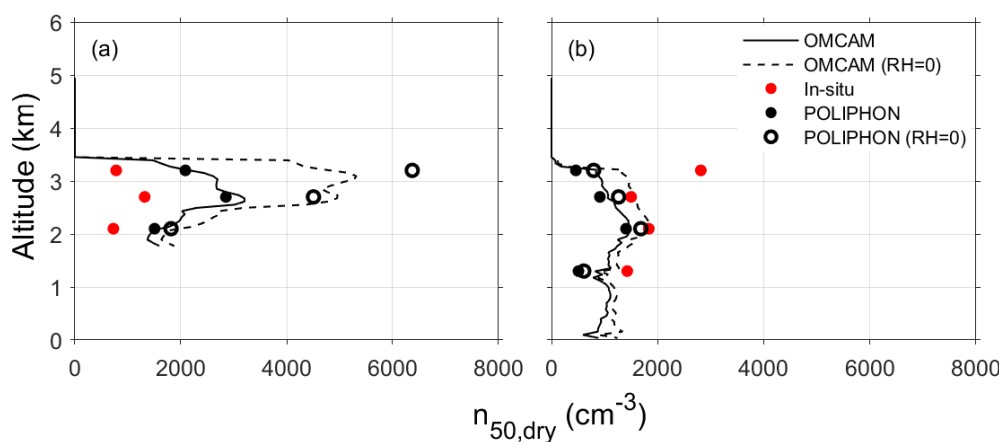

**Figure 7.** $n_{50,dry}$ concentrations estimated from CALIPSO satellite sata using OMCAM (solid line), POLIPHON (black dots) incorporated from Georgoulias et al. (2020), and in-situ aircraft observations (red dots) adopted from Tsekeri et al. (2017), over the land (a) and sea (b) surface close to Thessaloniki region. The dotted line and unfilled black circles represent the $n_{50,dry}$ estimated from OMCAM and POLIPHON, respectively, when hygroscopicity correction is not considered.





**Table 1.** POLIPHON conversion factors ($C$) and extinction exponents ($x$) for different aerosol subtypes.

| Type | $C$ | $x$ | Source |
|---|---|---|---|
| Dust | 8.855 | 0.7525 | Ansmann et al. (2019) |
| Continental | 25.3 | 0.94 | Mamouri and Ansmann (2016) |
| Marine | 7.2 | 0.85 | Mamouri and Ansmann (2016) |
| Smoke | 17 | 0.79 | Ansmann et al. (2021) |



**Table 2.** Bimodal lognormal volume size distribution parameters along with their limits considered in the sensitivity analysis. Abbreviations: VF - volume fraction, MR - mean radius, GSD - geometric standard deviation, CAM - CALIPSO aerosol microphysics.

| Aerosol subtype | Size distribution parameters | | | | | | | | | | | | | | |
| --- | --- | --- | --- | --- | --- | --- | --- | --- | --- | --- | --- | --- | --- | --- | --- |
| | VF fine | | | MR fine | | | MR coarse | | | GSD fine | | | GSD coarse | | |
| | CAM | min | max | CAM | min | max | CAM | min | max | CAM | min | max | CAM | min | max |
| Clean marine | 0.025 | 0.001 | 0.035 | 0.150 | 0.101 | 0.227 | 1.216 | 0.815 | 1.824 | 1.600 | 1.376 | 2.56 | 1.60 | 1.376 | 1.76 |
| Dust | 0.223 | 0.114 | 0.332 | 0.116 | 0.083 | 0.164 | 2.833 | 1.615 | 4.249 | 1.481 | 1.304 | 2.192 | 1.908 | 1.545 | 3.625 |
| Polluted cont. | 0.531 | 0.235 | 0.703 | 0.158 | 0.109 | 0.227 | 3.547 | 1.88 | 5.321 | 1.526 | 1.327 | 2.319 | 2.065 | 1.631 | 4.13 |
| Clean cont. | 0.050 | 0.001 | 0.069 | 0.206 | 0.136 | 0.310 | 2.633 | 1.501 | 3.950 | 1.61 | 1.385 | 2.592 | 1.899 | 1.538 | 3.589 |
| Elevated smoke | 0.329 | 0.168 | 0.49 | 0.144 | 0.098 | 0.211 | 3.726 | 1.938 | 5.589 | 1.562 | 1.359 | 2.437 | 2.143 | 1.671 | 4.285 |





**Table 3.** Percentiles of output $n_{j,dry}$ ensembles estimated from the sensitivity analysis relative to that of unperturbed case.

| Aerosol subtype | Percentiles of output $n_{j,dry}$ ensembles relative to unperturbed (%) | | | |
|---|---|---|---|---|
| | $5^{th}$ | $25^{th}$ | $75^{th}$ | $95^{th}$ |
| Clean marine | 0.044 | 35.8 | 119.25 | 194.05 |
| Dust | -41.30 | -28.54 | -0.68 | 34 |
| Polluted cont. | -56.26 | -20.77 | 88.15 | 259.38 |
| Clean cont. | -49.02 | 7.56 | 130.04 | 275 |
| Elevated smoke | -52.23 | -14.16 | 71.68 | 183.55 |





**Table 4.** $n_{50,dry}$ concentrations (in $cm^{-3}$) from in-situ measurements (Tsekeri et al., 2017) and CALIPSO measurements by using OMCAM and POLIPHON (Georgoulias et al., 2020) algorithms at different altitudes over land and sea regions around Thessaloniki, Greece. The values inside the bracket refers to zero humidity case (no hygroscopicity correction applied).

| Region | Altitude | In-situ | OMCAM (RH = 0) | POLIPHON (RH = 0) | CALIPSO - in-situ (%) | |
|---|---|---|---|---|---|---|
| | | | | | OMCAM (RH = 0) | POLIPHON (RH = 0) |
| **Land** | 2.1 | 727 | 1590 (1957) | 1504 (1816) | 119 (169) | 107 (150) |
| | 2.7 | 1318 | 3171 (5296) | 2851 (4505) | 141 (302) | 116 (242) |
| | 3.2 | 779 | 2160 (5401) | 2086 (6370) | 177 (593) | 168 (718) |
| **Sea** | 1.3 | 1427 | 826 (926) | 508 (609) | -42 (-35) | -64 (-57) |
| | 2.1 | 1834 | 1476 (1796) | 1405 (1683) | -20 (-2) | -23 (-8) |
| | 2.7 | 1601 | 1065 (1504) | 912 (1264) | -29 (0) | -39 (-16) |
| | 3.2 | 2814 | 841 (1357) | 459 (794) | -70 (-52) | -84 (-72) |