# Peer review of "Estimating cloud condensation nuclei concentrations from CALIPSO lidar measurements"

_Atmospheric Measurement Techniques, 2021_

## Author Response (AR1)

We thank all the reviewers for their time and effort in reviewing our manuscript. We found their comments to be very helpful in enhancing the quality of our article. Following are our point-by-point replies to the comments. Referee comments are given in black, our answers are given in blue.

**Reviewer-1:**

1. Page 1 / line 8: As a general comment, try to keep either present or past tense (but not both) in the text. For example, rephrase "...We have studied the sensitivity of the thus derived CCN concentration to the effect of variations of the initial size distributions..." to "...the sensitivity of the derived CCN concentrations to variations of the initial size distributions is also examined...".

Thank you for pointing this out. We have modified the sentence in the updated manuscript.

2. Page 1 / line 10: Similarly, rephrase "... We have also compared our results with the POLIPHON and found comparable results for extinction coefficients larger than 0.05 km-1..." to "...Our results are comparable to results obtained using the POLIPHON method for extinction coefficients larger than 0.05 km-1...".

Thank you for your suggestion. We have modified the sentence in the revised manuscript.

3. Page 1 / line 19: The IPCC reference should be corrected in the reference list. You may find here

(https://www.ipcc.ch/report/ar6/wg1/downloads/report/IPCC\_AR6\_WGI\_Citation.pdf) the proper way of citing the freshly released IPCC report.

Thank you for pointing this out. We have modified the citation in the revised paper.

**Comments 4-6:**

4. Page 2 / line 37: "Lidar sensors provide" instead of "lidar provides". Also, give a definition some lines before when the lidar word appears for the first time.

Thank you for the suggestion. We have changed the statement from "*lidar provides*…" to "*lidar measurements provide*…". The AMT readership is familiar with the acronym lidar. At least, it or its cousin radar are rarely introduced in AMT publications.

5. Page 2 / line 46: "a global" instead of "global".

Changed to "global 3D CCN and INP data sets".

6. Page 3 / line 75: "for altitudes between" instead of "in between altitudes". Changed.

7. Page 3 / line 79: Give a reference for GMAO.

We have added the reference for the GOES-5 model currently used in GMAO as follows:

"Molod, A., Takacs, L., Suarez, M., and Bacmeister, J.: Development of the GEOS-5 atmospheric general circulation model: evolution from MERRA to MERRA2, Geosci. Model Dev., 8, 1339–1356, https://doi.org/10.5194/gmd-8-1339-2015, 2015."

8. Page 3 /line 93: Please clarify the following sentence: "...Since the changes in lidar ratio from version 2 to version 4 are minor ( $\leq 1\%$ ) for all aerosol types except for clean continental (51%), we believe the aerosol models can still be used in our algorithm. However, for the case of clean continental aerosol subtype, further study is required to estimate the effect of change in lidar ratio on its microphysical properties. Having said that, we do not exclude it from our analysis for the completeness of our algorithm, leaving a scope of future validation study to examine its applicability in estimating the CCN concentrations from CALIPSO..."

It is not very clear what you want to stress here. Maybe some information is missing.

We understand the confusion and are sorry for not having been more clear. The lidar ratio used in the initial versions of the CALIPSO retrieval were provided in the CALIPSO aerosol model. However, the lidar ratios in the current version of the CALIPSO retrieval have been adjusted based on the findings from measurement campaigns. For some aerosol types, these values are no longer connected to the CALIPSO aerosol model. We wanted to emphasize that the changes in the lidar ratios used in the version 4 CALIPSO retrieval are mostly minor compared to earlier used values. We hence conclude that the aerosol microphysical properties from the CALIPSO aerosol model can still be used in our algorithm. To convey the same, we have added the following sentence (line 102-104) to the revised manuscript:

"Note that the lidar ratios used in version 4 of the CALIPSO retrieval have been adjusted from earlier versions based on the findings from atmospheric measurements (Kim et al., 2018) and don't necessarily connect to the CALIPSO aerosol model."

9. The methodology is well explained. The same stands for the sensitivity analysis.

Thank you very much for the positive feedback.

10. Page 10 / line 299: "The ratio between the CCN concentrations estimated using POLIPHON (CCNPOLI) and OMCAM (CCNOMCAM) algorithms for varying extinction coefficients at a supersaturation of 0.15% is shown in Figure 5.". The values compared here are not from G20 but were calculated on an aerosol-type basis from the authors on their own. If I am correct, then the values appearing in Fig. 5 are not RH corrected (RH=0). Please clarify this in the revised manuscript if this is the case.

Thank you for pointing this out. We have modified the sentence in the updated manuscript as follows.

"The ratio between the CCN concentrations estimated using POLIPHON (CCNPOLI) and OMCAM (CCNOMCAM) algorithms for varying extinction coefficients at a supersaturation of 0.15% and zero relative humidity is shown in Figure 5."

11. Page 10 / line 310: "This may be either because of different instruments or sample size considered to derive the size distributions used in both the algorithms..." should be rephrased to "This may be due to the different approaches followed and sample size considered to derive the size distributions used in the two algorithms."

Thank you for the suggestion. We have rephrased the statement in the revised manuscript.

12. Page 11 / line 316: Please stress that we cannot still say which algorithm performs better. This is why there should be a detailed evaluation of both the algorithms in the future. Is there any advantage/disadvantage that would make any of those two algorithms preferable (e.g. one being faster / or more detailed by means of physics compared to the other, allow direct correction for RH, etc.)? It would be nice to add a couple of lines here.

Thank you for pointing this out. We have already stressed the need for a detailed evaluation of both algorithms (POLIPHON and OMCAM) at the end of Section 4.3. We have also added the following sentences to the updated manuscript to weigh the theoretical advantages and disadvantages of both the algorithms at the end of Section 4.2:

When it comes to ease of application, the POLIPHON method with its simple extinction-to-CCN conversion is more straightforward while the OMCAM algorithm – at the present stage – is more complex and computationally expensive. Despite the complexities, OMCAM incorporates a hygroscopicity correction methodology which is essential for a CALIPSObased CCN retrieval (Georgoulias et al., 2020). Furthermore, the computation time in the OMCAM algorithm can be drastically reduced by either (i) parameterizing the output CCN concentrations in terms of the type-specific extinction coefficient and RH values or (ii) creating a look-up table of CCN concentrations at different extinction coefficients and RH values for different aerosol subtypes. However, such developments are not within the scope of the present work which focuses on the theoretical description of the OMCAM algorithm.

13. Page 12 / line 365: Replace "such huge difference" with "the large discrepancy".

Thank you for your suggestion. We have modified it accordingly.

14. Page 12 / line 379: I suggest removing POLIPHON from the phrase "...Utilizing the aerosol type-specific CCN parameterizations from the POLIPHON method (Mamouri and Ansmann, 2016)...". You may write "Utilizing the aerosol type-specific CCN parameterizations from Mamouri and Ansmann (2016)...".

Thank you for pointing it out. We have modified it in the revised manuscript.

15. Page 13 / line 411: You may rephrase "...We have described a novel methodology to retrieve cloud relevant CCN concentrations from CALIPSO measurements illustrating the potential of CALIPSO..." to 'Following the first CALIPSO CCN retrievals from Georgoulias et al. (2020) with the POLIPHON algorithm, in this work we suggest a novel methodology to retrieve cloud relevant CCN concentrations from CALIPSO measurements further illustrating the potential of CALIPSO..."

Thank you for your suggestion. However, we believe that the concluding paragraph of our manuscript should focus only on the present and future aspect of our work. Nevertheless, we have added the following sentence in the Introduction section of our manuscript (line 51-54) to highlight the first CALIPSO CCN retrieval from Georgoulias et al. (2020).

"Georgoulias et al. (2020) for the first time estimated CCN concentrations from CALIPSO measurements by using the POLIPHON technique and found good agreement with the coincident airborne in-situ measurements taken during the ACEMED-EUFAR (evaluation of CALIPSO's aerosol classification scheme over Eastern Mediterranean) campaign (Tsekeri et al., 2017)."

**Reviewer-2:**

1. As mentioned in the general comment, after L48, it should be added another paragraph to state the motivation of proposing this new method for estimating CCN concentrations, which is different from the widely used POLIPHON method. To my point of view, the potential advantage for the proposed method is not necessary for calculating those region-varied conversion factors to obtain  $n_{j,dry}$  (Ansmann et al. 2019AMT, He et al., 2021AMT), which cannot be obtained in the regions without sun photometer observations. Instead, CAMel and MOPSMAP package can solve this problem, making the global estimation of CCN concentrations possible. If so, add a new paragraph to state that.

We appreciate the reviewer's comment. It is crucial to note that the size distributions and refractive indices included in the CALIPSO aerosol model are derived from cluster analysis of AERONET measurements and from independent measurement campaigns. In either case, the site-specific measurements limit our coverage of different regions of our globe. However, the cluster analysis of long-term AERONET measurements already has provided us with deeper insights into the major global aerosol types and their consistent microphysical properties across different measurement sites (Omar et al., 2005). Having said that, we still believe this consistency will be disrupted in real-atmosphere conditions of complex aerosol mixtures. This circumstance motivated us to perform a sensitivity analysis and to quantify the effect of variations of the size distributions in our algorithm on the derived CCN concentrations. Also, the POLIPHON method has not yet been applied to long-term CALIPSO data for studying aerosol-cloud interactions. This is our long-term goal and to achieve it, we wanted to develop a new methodology that is self-consistent within the CALIPSO framework. To emphasize this circumstance, we have added the following sentences to the Introduction:

"The approach for retrieving cloud-relevant aerosol microphysical properties has not yet been implemented for spaceborne lidar measurements. This study, therefore, presents a new methodology for obtaining height-resolved aerosol number concentrations from CALIPSO measurements within the CALIPSO framework, i.e. without relying on externally inferred conversion factors."

2. The basic information of the data set used in subsection 4.1 and 4.2 are absent, such as the location (global or a few sites) and period (a few months or year) of the selected data. This may confuse the readers, especially considering the large differences of conversion factors in the POLIPHON method. It is suggested to show some cases (after figure 5) that compare the CCN concentration profiles for five aerosol types with POLIPHON and OMCAM algorithms.

We are sorry for the confusion. The sensitivity tests (Section 4.1) and theoretical comparison between OMCAM and POLIPHON methods (Section 4.2) are performed solely based on synthetic data. As stated in lines 239, 269, and 270 of the revised manuscript, while performing the sensitivity experiments, we use a preset extinction coefficient value of 0.1 km-1 and zero relative humidity, and vary the input size distribution parameters (mean radii, standard deviations, and volume fraction) of our algorithm. Coming to Section 4.2, we preferred the theoretical comparison of both the algorithms because it is hard to find real world scenarios with only one aerosol subtype and thus using real data can skew the results. We have added the following sentences (lines 307-310) for further clarification in the revised paper.

"In this section, we present a theoretical comparison of the CCN concentrations estimated using the OMCAM and POLIPHON methods (Mamouri and Ansmann, 2016). Both algorithms' primary input is the aerosol type-specific extinction coefficient. Hence, we consider a range of extinction coefficients and compute the corresponding theoretical CCN concentrations with both algorithms. To estimate CCN concentrations with POLIPHON, we use the extinction-to-CCN conversions given in Eq. (1)."

3. The difference (or important improvement) between this method and Georgoulias et al. (2020) should also be stated.

Thank you for pointing this out. We have added the following sentences in our revised manuscript (lines 331-338):

"When it comes to ease of application, the POLIPHON method with its simple extinction-to-CCN conversion is more straightforward while the OMCAM algorithm – at the present stage – is more complex and computationally expensive. Despite of the complexities, OMCAM incorporates a hygroscopicity correction methodology which is essential for a CALIPSObased CCN retrieval (Georgoulias et al., 2020). Furthermore, the computation time in the OMCAM algorithm can be drastically reduced by either (i) parameterizing the output CCN concentrations in terms of the type-specific extinction coefficient and RH values or (ii) creating a look-up table of CCN concentrations at different extinction coefficient and RH values for different aerosol subtypes. However, such developments are not within the scope of the present work which focuses on the theoretical description of the OMCAM algorithm."

4. L26-27: "Quaas et al., 2008; Quaas et al., 2020" -> 'Quaas et al., 2008, 2020'

Thank you for identifying this mistake. We have rectified it in the updated manuscript.

5. L40:  $n_{280}$  has been modified to  $n_{250}$  in the following papers (Mamouri and Ansmann, 2016ACP; Ansmann et al., 2019ACP).

Thank you for pointing this out. We have changed  $n_{280}$  to  $n_{250}$  in the updated manuscript.

6. L76: volume or particle depolarization ratio? It should be clearly stated.

Thank you for pointing this out. We use the particle depolarization ratio and have stated it in the updated manuscript.

7. L92-93: What if CALIPSO level-2 aerosol subtype is misclassified (Ansmann et al., 2021 FENVS)? It should be mentioned and added some sentences to evaluate the related impact.

Thank you for your suggestion. We have added the following sentence (lines 299-300) to our manuscript along with the reference.

*"Furthermore, aerosol misclassification in the CALIPSO aerosol-typing scheme (Ansmann et al., 2021a) may introduce errors in the OMCAM algorithm."*

8. L103: 'spheroids' -> 'non-spheroids'? please confirm.

We apologize for the confusion. The MOPSMAP package uses combination of the T-matrix method and improved geometric optics method for spheroid aerosols.

9. L116: define the  $\alpha$  in equation (1) here. *x* is aerosol extinction exponent. The unit should be given for each parameter in equation (1). Besides, you should also cite when giving equation (1) (Shinozuka et al., 2015ACP).

Thank you for pointing it out. The parameter " $\alpha$ " was already defined before Eq. (1) in line 124 of the updated manuscript. We have added units to all parameters and now cite the article in our updated manuscript.

10. L124: Conversion factors vary from region to region. Therefore, as suggested before, it is better to choose a specific site (maybe the site in Limassol, where many target aerosol types can be observed) and give CCN profile comparison between POLIPHON and OMCAM algorithms.

As explained in the response to the second comment, we have performed a theoretical comparison between OMACM and POLIPHON retrievals which is much more universal than the Referee's suggestion. As the POLIPHON conversion factors may vary with regions, we use their averaged values suggested by Ansmann et al. (2019, 2021) for the application to CALIPSO data (already stated in lines 135-136).

11. L127-134: This paragraph is logically confusing and should be rephrased to introduce the general content of section 3.

We apologize for the confusion. We have modified the paragraph in the revised manuscript as:

"This section describes the algorithm used in the present work to derive CCN concentrations from the CALIPSO profiles of extinction coefficient, backscatter coefficient, depolarization ratio, and aerosol subtype information. We begin with the scaling procedure of the normalized size distributions from the CAMel to obtain the actual aerosol size distribution. After that, we explain the hygroscopicity correction followed by the CCN parametrization adopted in our algorithm. Finally, we discuss the application of the CCN retrieval algorithm to CALIPSO level 2 aerosol profile data."

12. L232: How can the conclusion 'fine mode as the primary contributor to the output aerosol number concentration' be drawn from figure 2?

As discussed in the same paragraph, from Figure 2, we see that the fine mode size distribution parameters such as mean radius and standard deviations have the maximum impact on the output  $n_{j,dry}$  concentrations of up to 600% and 100%, respectively.

13. L312: Rephrase this sentence to more clearly state that conversion factors for marine aerosols are estimated from the Barbados AERONET site?

We have rephrased the sentence in the revised manuscript as:

"The POLIPHON conversion factor for marine aerosol is estimated from 7.5 years of measurements between 2007 and 2015 at the Barbados AERONET site (Mamouri and Ansmann 2016)."

14. L135 and Figure 5: There is a systematical factor of around 3-5 for marine aerosol types between the CCN concentration from two different methods. It would be better to try to give a more specific discussion, especially considering the proposed method is expected to extend to a global scale.

We understand reviewer's point here. We have addressed the possible causes behind the differences in lines 323-331 of the revised manuscript as given below. In short, we believe

that this may be a result of different approaches and sample sizes used to derive the size distributions in both the methods. Furthermore, we stress the need for a detailed validation study to identify which algorithm is more accurate in estimating CCN concentrations for marine aerosols.

"This may be due to the different approaches followed and sample sizes considered to derive the size distributions used in the two algorithms. The POLIPHON conversion factor for marine aerosol is estimated from 7.5 years of measurements between 2007 and 2015 at the Barbados AERONET site (Mamouri and Ansmann, 2016). In contrast, the marine model used in OMCAM is derived from in-situ measurements of sea-salt size distributions produced from breaking waves, taken during the SEAS experiment at Bellows Air Force Station, Oahu, Hawaii between 21 and 30 April 2000. Studies found that the AERONET size distributions can be significantly different from the in-situ measurements – especially under high relative humidity conditions (Chauvigne et al., 2016; Schafer et al., 2019). Further studies involving type-specific comparisons of both the aerosol number concentrations and the CCN concentrations with in-situ measurements are required to test the reliability of both algorithms (Mamali et al., 2018)."

**15. Figure 6: The trajectory of the sub-satellite point should be added.**

Thank you for your suggestion. It is aimed at better placing the location of the measurements. We have tried adding an inset of the satellite track but were not satisfied with the results. Therefore, we kept the figure as it is and revised the caption for a better description of the location of the measurement to "for a CALIPSO overpass over the Thessaloniki region of northern Greece on 9 September 2011."

16. Figure 7: It is great to see that CCN concentration profiles from POLIPHON and OMCAM are consistent with each other after hygroscopicity correction very well. It is expected to conduct more comparison with in-situ measurement in the future as reported in (Mamali et al., 2019AMT).

We have added the suggested reference in the revised manuscript.